# Social stratification reflected in bone mineral density and stature: Spectral imaging and osteoarchaeological findings from medieval Norway

Elin T. Brødholt[1,2]*, Kaare M. Gautvik[3], Clara-Cecilie Günther[4], Torstein Sjøvold[5], Per Holck[1]

**1** Department of Molecular Medicine, Institute of Basic Medical Sciences, University of Oslo, Oslo, Norway, **2** Department of Forensic Sciences, Oslo University Hospital, Oslo, Norway, **3** Unger-Vetlesen Institute, Lovisenberg Diaconal Hospital, Oslo, Norway, **4** Norwegian Computing Center, Oslo, Norway, **5** Stockholm University, Stockholm, Sweden

* e.t.brodholt@medisin.uio.no

**Data Availability Statement:** All relevant data are within the paper and its Supporting Information files.

## Abstract

This study presents skeletal material from five medieval burial sites in Eastern Norway, confined to one royal burial church, one Dominican monastery, and three burial sites representing parish populations. We combine osteological analysis and Dual Energy X-Ray Absorptiometry, studying the remains of 227 individuals (102 females and 125 males) employing young, middle, and old adult age categories. The aim is to assess bone mineral density as a skeletal indicator of socioeconomic status including stature as a variable. We detected that socioeconomic status significantly affected bone mineral density and stature. Individuals of high status had higher bone mineral density ($0.07$ g/cm$^2$, p = 0.003) and taller stature (1.85 cm, p = 0.017) than individuals from the parish population. We detected no significant relationship between young adult bone mineral density and socioeconomic status (p = 0.127 and 0.059 for females and males, respectively). For males, high young adult bone mineral density and stature varied concordantly in both status groups. In contrast, females of high status were significantly taller than females in the parish population (p = 0.011). Our findings indicate quite different conditions during growth and puberty for the two groups of females. The age-related pattern of bone variation also portrayed quite different trajectories for the two socioeconomic status groups of both sexes. We discuss sociocultural practices (living conditions during childhood and puberty, as well as nutritional and lifestyle factors in adult life), possibly explaining the differences in bone mineral density between the high-status and parish population groups. The observation of greater differences in bone mineral density and stature for females than males in the medieval society of Norway is also further discussed.

**Funding:** The author(s) received no specific funding for this work.

**Competing interests:** The authors have declared that no competing interests exist.

## Introduction

Throughout modern history, socioeconomic status (SES) has been linked to health and longevity [1, 2], impacting rates of mortality and morbidity for almost any disease and condition [3]. Higher economic status usually implies access to healthier living conditions, improved sanitation, and better health care [2]. The medieval period in Norway was characterized by marked social stratification and poverty, which was believed to be hereditary. The sustained poverty endured by generations of people from the lower social strata led to deficient nutrition, poor housing conditions, and inadequate hygiene, resulting in lower stature and increased vulnerability to certain diseases [4].

The development of an individual's maximal bone mass (peak bone mass) is 60–80% genetically determined [5, 6] but is modified by both pre- and post-natal determinants, e.g., nutrition, vitamin supply, and presence of chronic diseases. The most rapid skeletal growth occurs within two years after birth, and a second skeletal growth burst corresponds to puberty. The increase in bone mass during puberty is greater in boys than in girls due to a more prolonged period of accelerated growth [7, 8]. Thus, our age-dependent bone mass is influenced by several variables such as sex, nutrition, endocrine factors, mechanical strain, disease, and exposure to risk factors, while peak bone mass is predominantly genetically determined [8, 9]. Suboptimal lifestyle factors such as insufficient nutrition and lack of physical activity, especially between the ages of 10 to 18, may result in reduced bone mass and strength in adulthood [6, 7]. Other known modifiable risk factors for low bone mass include low calcium intake, tobacco smoking, excessive alcohol intake, lack of weight-bearing exercise, and sunlight exposure [10].

The high occurrence of osteoporosis in modern Caucasian populations led to numerous studies focusing on this condition in archaeological skeletal remains [11–20], searching to unveil if patterns of bone loss in the past resembled those observed in the modern population. The initial assumption was that the lifestyles associated with past populations would be expected to lower the risk of osteoporosis [21]. Instead, these studies demonstrated a varied pattern of age- and sex-related bone loss: ranging from less to generally similar or greater bone loss than in modern populations. Furthermore, recent research on archaeological skeletal material emphasizes that the risk factors associated with osteoporosis are multiple, complex, and often symbiotic [22].

Bone mineral density (BMD) has varied notably between archaeological populations and time periods in Scandinavia [4, 12–15, 17, 23, 24]. The age- and sex-related BMD and patterns of bone loss in these populations are diverse, and the findings demonstrate the lack of consistent trends. It has been unclear to what extent this variation in BMD can be linked to social inequality. Clinical studies [25–27] demonstrate the effect of SES on skeletal BMD; however, they conclude that it is difficult to establish a consistent and conclusive positive or negative association. Education and income are positively associated with BMD but depend on factors such as sex, age, and ethnicity. Research on BMD variation, as measured by Dual Energy X-Ray Absorptiometry (DXA), related to SES in archaeological skeletal material has been limited [4–6]. However, the findings demonstrate the complex and not so straightforward relationship between SES, sex, and lifestyle factors such as nutrition and physical activity. For example, Di Stefano, Boano, Rabino Massa, Isaia and Panattoni [28] and Borre, Boano, Di Stefano, Castiglione, Ciccone, Isaia et al. [29] found that high-status individuals buried inside the San Michele Church, North-West Italy, had significantly lower BMD than individuals buried in the cemetery. A larger intake of dairy products, more sun exposure, and greater physical activity in the cemetery group were presumed to explain the observed pattern. Zaki, Hussien and El Banna [30] investigated the occurrence of osteoporosis in two social classes from the Old Kingdom in Giza and found a higher prevalence in male workers compared to male high

officials and female high officials versus female workers. The authors related the findings to nutritional stress and increased workload in male workers, and a sedentary lifestyle among female high officials.

Recently, we [23] showed that differences in SES were reflected in BMD variation as the parish population had significantly lower BMD than individuals of high SES. The present study complements and extends our previous investigation [23] and aims to analyze if and how BMD can be used as a skeletal indicator of SES confined to the medieval period in Norway (1030–1536 AD). Research has shown that stature reflects SES strongly and consistently, and this positive association is widely observed across cultures [26, 31]. Stature is among the most heritable traits in humans, but differences in stature between SES groups have shown that it is modified by environmental factors. These factors are particularly important during development [32–35]. Furthermore, research has revealed that the crucial periods for bone mineralization in non-adults overlap with those for stature attainment [26]. We, therefore, implemented stature as a variable in our analyses. Based on previous research by the authors [23], we have classified the five burial sites included in the study as essentially reflecting two SES groups. We present the first in-depth mapping of BMD variation in medieval Norway and discuss how socioeconomic factors may have contributed to the results.

## Materials

The skeletal material from the five medieval burial sites included in our study (Fig 1) spans from the 11th to the 16th century AD in Norway. It constitutes part of the Schreiner Collection at the Division of Anatomy, University of Oslo. The skeletal remains from these sites were previously analyzed with DXA as part of a larger multi-period mapping of BMD variation in Norway [23].

The Church of St. Mary (c. 1050–1540 AD, Fig 1) was a burial place for the Norwegian royal family [36], in addition to the nobility and clergy [37], as well as members of the aristocracy and gentry that bought a burial plot, or paid for upkeep in old age (*corrody*), having prayers said and sins absolved, etc. [37, 38]. Different aspects of the skeletal material have been analyzed [4, 39–43]. Previous research undertaken by Brødholt and Holck [39] revealed a high incidence of skeletal trauma at this burial site, presumably linked to the civil wars in Norway during the 12[th] and 13[th] centuries. Compared to other royal burial churches in Europe, the Church of St. Mary was accessible to a wide range of people connected to the royal family, even to people from other SES groups, thus creating a broad basis for legitimacy [37]. However, the common denominator was their high SES in medieval society.

St. Olav's Monastery was founded in 1239 AD [44–46] and was presumably operative until the reformation in 1537 AD when the mendicant friars were forced to leave Denmark-Norway [47, 48]. The Order of Preachers (O.P.) was an extrovert order with many versatile tasks in medieval society. Besides preaching and pastoral care, they acted as emissaries for the King and bishop, witnesses, and independent tradesmen [46]. The intellectual mark of the order earned them central positions within administration and diplomacy [49]. They were recruited from the gentry and bourgeoisie, as well as from intellectual contexts, such as universities [50]. Initially, the friars were bound by a vow of poverty, renouncing property and relying on begging and charity [47, 48]. However, they were accused of leaving this ideal in favor of a more material and luxurious existence accepting gifts and regular income from property [46]. There were indications that both friars and other citizens were buried at this site [48, 51]. The skeletal material included in this study was excavated from the south wing in 1924–26 [48], and the remains of 23 individuals were unearthed [44]. This material was first analyzed by Wagner [52] and later by Holck [4].

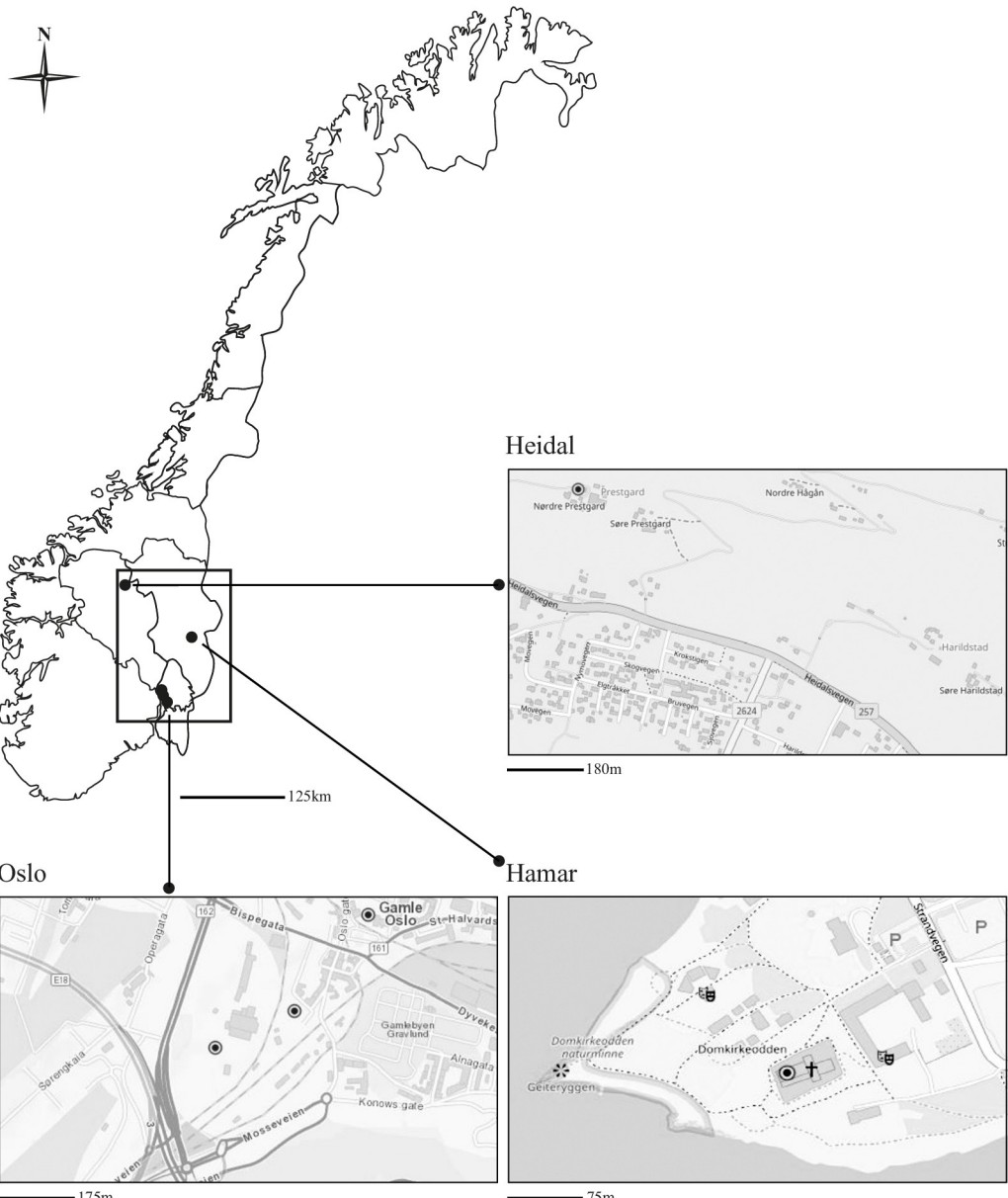

**Fig 1. Map showing the location of the five burial sites included in the study.** 1. Church of St. Mary (ca. 1050–1540 AD); 2. Church of St. Clemens (980–1030 AD); 3. St. Olav's Monastery (1239–1537 AD); 4. Church of Prestgard (early 11th century-1531 AD); 5. Hamar Cathedral (early 12th 77 century-1565/1570 AD). Figure created with map data from U.S. Geological Survey.

The Church of St. Clemens (Fig 1) was a parish church that operated throughout the entire medieval period [36] and was established in the first half of the 11[th] century AD [53]. The whole church ruin was excavated in 1920–21, thus unearthing a large number of burials. The skeletal material was partly analyzed by Wagner [52] in the 1920s and its entirety in 2001 by Holck and Kvaal [54]. The latter authors also included a smaller skeletal material in their analysis, excavated from a confined area inside the church in 1970–71 [55]. Overall, they estimated that the skeletal material from this site constituted a minimum number of 998 individuals.

The Church of Prestgard (Prestgardskyrkja) (Fig 1) was a stave church built during the first half of the 11<sup>th</sup> century in the mountain valley of Heidal, in southern inland Norway. The churchyard connected to this church was thought to have served the small parish community [56] and, judging by the size of the church, most likely represented a local circuit of farmers [57]. In the 1340s, there were 25 farms in this village, entailing ca. 200–250 people [58]. The remains of 77 individuals and additional comingled remains were unearthed in 1925 when part of the churchyard at Nørdre Prestgard was excavated. The medieval burials at this cemetery were believed to directly follow pre-Christian graves [59], while some skeletons found in a common grave were thought to be the result of the Black Death [56, 58, 60]. The skeletal material from this site was later analyzed by Holck [4].

Hamar Cathedral (Fig 1), originally a Romanesque basilica, was close to the small market town of Hamar, the fifth episcopal seat in Norway. The church functioned from the beginning of the 12<sup>th</sup> century to c. 1567 AD, when the church was irrevocably damaged by fire [61]. In 1991–92, remains from more than 500 individuals were unearthed when a protective glass building was constructed over the church ruins [61, 62]. The skeletal material was analyzed by Sellevold [61], who debated the role of the cathedral cemetery. The burial pattern was complex and indicated that the church's role as a burial church probably encompassed several socioeconomic strata from the inhabitants of the large diocese of Hamar to the ecclesiastical community as well as the aristocracy and gentry.

## Methods

Ethical approval was obtained from the National Committee for Research Ethics on Human Remains, The Norwegian National Research Ethics Committees (2015/396 and 2016/304). Furthermore, the authors obtained all necessary permits for the described study, which complied with all relevant regulations.

### Osteological analysis

The Schreiner Collection database and archive literature provided osteological data on the skeletal materials, as did various publications: St. Olav's Monastery [51, 52], The Church of Prestgard [59], Hamar Cathedral [61], the Church of St. Mary [39–42] and the Church of St. Clemens [54]. All skeletal remains were subject to a separate evaluation of sex, age at death, stature, pathology, and trauma. We conducted this evaluation according to traditional methods given by Buikstra and Ubelaker [63]; sexing of crania [64], assessment of pelvic features [65], estimation of age-at-death was performed by evaluating suture closure [66] as well as the pubic symphysis [67, 68]. We estimated stature according to Trotter and Gleser [69], Trotter and Gleser [70], and documented pathology and trauma following Ortner [71] and Aufderheide and Rodríguez-Martín [72]. We applied broad age groups: Young Adult (20–35 years), Middle Adult (35–50 years), and Old Adult (50+), as per Buikstra and Ubelaker [63]. The skeletal material was classified into two proposed socioeconomic groups: 1. High-status burials and 2. Parish population burials. The Church of St. Clemens, The Church of Prestgard, and Hamar Cathedral were classified in the latter group, while the burials at the Church of St. Mary and St. Olav's Monastery were classified as high status.

### Dual-Energy X-Ray Absorptiometry

A Lunar iDXA (GE Healthcare Lunar, Madison, WI, USA) was used for the BMD measurements, and the femur neck (*collum femoris*) was defined as the region of interest. A combination of water and plastic boards was used as a soft-tissue substitute, and the femur bone was positioned between the boards and the water within a specially constructed frame. The femur

was positioned with the anterior surface facing up, with the neck in a horizontal plane and the diaphysis oriented parallel to the scanner's axis. Each femur was scanned three times, and both femora from one individual were measured, if available. A quality assurance (QA) procedure was performed daily, in addition to a phantom used as a separate control measure. For extensive details on the procedure, inclusion criteria, and cross-calibration, see [23]. The study adopted BMD data from the DXA analysis by Holck [4], and we included 71 individuals with scans in accordance with the DXA analysis criteria. In addition, these BMD values were cross-calibrated to reduce any systematic differences in average BMD measurements between scanners (see [23] for Bland-Altman plot and further details).

## Statistical analysis

The variation in neck mean BMD and stature were modeled using linear regression. The distributions of neck mean BMD and stature were first visually inspected by histograms and QQ-plots and found to be approximately normally distributed. With neck mean BMD as the response variable, age group (Young, Middle, and Old Adult), sex, and SES (High-status burials and Parish population burials) were used as explanatory variables. When stature was the response variable, sex and SES group were the explanatory variables. Two sample t-tests were applied to compare the mean stature in the two SES groups and the mean BMD between SES groups for a specific age group for females or males. Model assumptions of independence and normality were checked using residual plots. The analyses were conducted in R [73], and the boxplots were created using the package ggplot2 [74].

## Results

### Osteological analysis

Age groups and sex distribution for each burial site included in the study are shown in Table 1, presenting 227 individuals in total (102 females and 125 males) after the exclusion of 20 individuals due to bone pathology and/or trauma to weight-bearing bones. Seventy-six individuals represent the high-status group, and 151 represent the parish population group. The estimated mean stature, SD, and range for each sex and SES group are shown in Table 2. The details are included in the S1 Table. Men had a significantly higher mean stature than females when adjusting for SES group (13.1 cm, p<0.001, linear regression model, Table 3). SES significantly affected stature as individuals of high status, on average, were 1.8 cm taller than individuals from the corresponding parish population (p = 0.017, linear regression model, Table 3). The mean statures in the high-status group were 175.4 cm (SD 5.0) for males and 164.2 cm (SD 5.9) for females, while the mean statures in the parish group were 174.6 cm (SD 5.8) for males and 160.7 cm (SD 4.9) for females (Table 2). When considering the data for each sex separately, females in the high-status group were significantly taller than females in the parish population group (3.5 cm, p = 0.011, two sample t-test, Table 4). No statistical difference was detected for males (0.8 cm, p = 0.410). There were no significant differences in stature within the SES groups.

### DXA-analysis

The age-related variation of femur neck mean BMD for both sexes in the SES groups is shown in Fig 2 and Table 5, and the changes are given as percent of the mean BMD in the Young Adult category. The details are included in the S1 Table. In addition, we compared the age-related variation to those found in a modern reference population from USA/Northern Europe (Table 5). In the medieval period, men had a significantly higher mean BMD than women,

**Table 1. Age groups and sex distribution for each burial site.**

| Burial site, County | SES | Sex | Age | | | n |
|---|---|---|---|---|---|---|
| | | | YA | MA | OA | |
| Church of St. Mary, Oslo | High | Female | 7 | 7 | 3 | 17 |
| | | Male | 11 | 15 | 12 | 38 |
| | | Total | | | | 55 |
| St. Olav's Monastery, Oslo* | High | Female | 5 | 2 | 2 | 9 |
| | | Male | 5 | 4 | 3 | 12 |
| | | Total | | | | 21 |
| **Total high status sample** | | Female | 12 | 9 | 5 | 26 |
| | | Male | 16 | 19 | 15 | 50 |
| | | Total | | | | 76 |
| St. Clemens Church, Oslo* | Parish | Female | 4 | 2 | 4 | 10 |
| | | Male | 5 | 2 | 2 | 9 |
| | | Total | | | | 19 |
| Prestgardskirken, Innlandet* | Parish | Female | 2 | 5 | 8 | 15 |
| | | Male | 2 | 4 | 10 | 16 |
| | | Total | | | | 31 |
| Hamar Cathedral, Innlandet | Parish | Female | 9 | 14 | 28 | 51 |
| | | Male | 23 | 15 | 12 | 50 |
| | | Total | | | | 101 |
| **Total parish sample** | | Female | 15 | 21 | 40 | 76 |
| | | Male | 30 | 21 | 24 | 75 |
| | | Total | | | | 151 |
| **Total study sample** | | **Female** | **27** | **30** | **45** | **102** |
| | | **Male** | **46** | **40** | **39** | **125** |
| | | **Total** | | | | **227** |

* Data from Holck (2007).

YA: Young Adults (20–35 years), MA: Middle Adults (35–50 years) and, OA: Old Adults (> 50 years). F: females and M: males. Table modified from Brødholt, Günther, Gautvik, Sjøvold and Holck [23].

given the same age-at-death and SES (0.10 g/cm$^2$, p<0.001, linear regression model, Table 3), and mean BMD was significantly lower in middle adulthood (-0.11 g/cm$^2$, p<0.001) and old adulthood (-0.17 g/cm$^2$, p<0.001) compared to young adults, given same sex and SES.

Distinct SES had a significantly affected BMD; individuals from the parish population had lower BMD than individuals of high status (-0.07 g/cm$^2$, p = 0.003). In addition, mean BMD was compared between SES groups for a specific age group and showed that old adult females in the high-status group had significantly higher mean BMD than the old adult females in the

**Table 2. Estimated mean stature (cm), SD and range for each sex and SES group.**

| Socioec. group | Sex | n | mean | SD | min | max |
|---|---|---|---|---|---|---|
| High status | Females | 25 | 164.2 | 5.9 | 151 | 173 |
| | Males | 50 | 175.4 | 5.0 | 164 | 186 |
| Parish population | Females | 76 | 160.7 | 4.9 | 151 | 173 |
| | Males | 76 | 174.6 | 5.8 | 161 | 190 |

F: females and M: males.

**Table 3. Results of the linear regression model for BMD and stature.** Estimated effect with females as reference level.

| | Estimated effect | p-value |
|---|---|---|
| **BMD** | | |
| Intercept | 1.07 | p<0.001 |
| Male | 0.10 | p<0.001 |
| Middle Adult* | -0.11 | p<0.001 |
| Old Adult* | -0.17 | p<0.001 |
| Parish pop. | -0.07 | 0.003 |
| **Stature** | | p<0.001 |
| Intercept | 162.94 | p<0.001 |
| Male | 13.12 | p<0.001 |
| Parish pop. | -1.85 | 0.017 |

*Estimated effect (g/cm$^2$) with young adulthood as reference level, given same sex and SES.

parish population group (p = 0.011, two sample t-test, Table 4). However, we detected no significant difference in males or within the two SES groups.

**Young adult mean BMD.** Overall, individuals in the high-status group displayed a significantly higher mean BMD than individuals in the parish group (0.07 g/cm$^2$, p = 0.003, linear regression model, Table 3). When we studied the young adult category, the mean BMD did not differ significantly between the high-status and parish groups. The young adult mean BMD for high-status females was 1.094 g/cm$^2$, while the corresponding estimate for high-status males was 1.185 g/cm$^2$. The young adult mean BMD values for males and females in the parish population were somewhat lower (1.080 and 1.014 g/cm$^2$, respectively). However, the difference did not reach significance (p = 0.059 and 0.127, respectively, Table 4). Compared to modern reference levels (Table 5), young adult females in the medieval period had higher mean BMD (p = 0.02, one sample t-test), as previously shown in [23]. However, this result did not reach significance after adjusting for multiple testing (q-value 0.13, Benjamini-Hochberg procedure). The mean BMD values for young adult males in the medieval period did not differ significantly from males in the modern reference population (1.080 g/cm$^2$) [23].

**Age-related BMD variation.** The female age-related BMD reduction in the high-status group was characterized by marked bone loss from young to middle adulthood (from 1.094 to 0.898 g/cm$^2$, - 17.9%). Afterwards, BMD increased from middle to old adulthood (from 0.898

**Table 4. Results of the two sample t-tests for stature and BMD comparisons.**

| | t-value | p-value |
|---|---|---|
| **Two sample t-test stature** | | |
| Females High status vs. Parish | 2.70 | 0.011 |
| Males High status vs. Parish | 0.83 | 0.410 |
| **Two sample t-test BMD** | | |
| Females YA, High status vs. Parish | 1.59 | 0.127 |
| Females MA, High status vs. Parish | -0.06 | 0.951 |
| Females OA, High status vs. Parish | 3.62 | 0.011 |
| Males YA, High status vs. Parish | 1.98 | 0.059 |
| Males MA, High status vs. Parish | 0.29 | 0.771 |
| Males OA, High status vs. Parish | 1.29 | 0.208 |

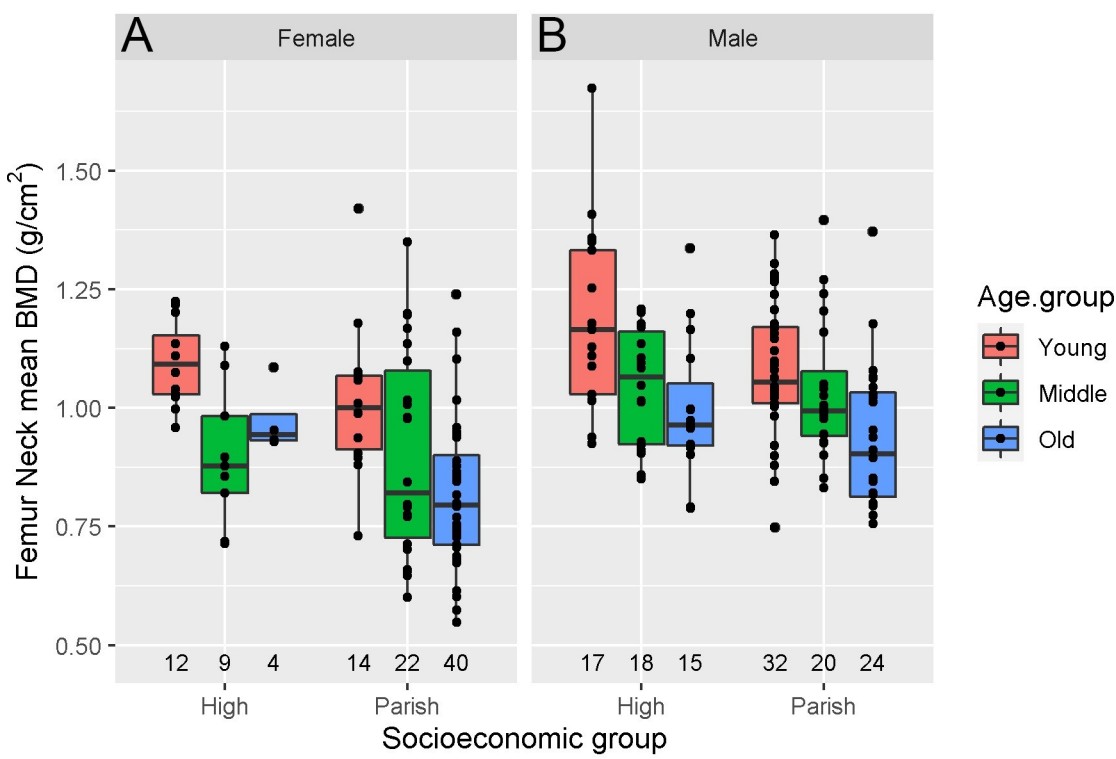

**Fig 2.** Femur neck mean BMD at the five medieval burial sites: females (A) and males (B). Young, Middle, and Old Adult age groups. The number of individuals in each age group is stated under each box.

to 0.975 g/cm², 7%). The early bone loss among the high-status females was markedly greater than that displayed by the parish population females, which experienced a reduction in BMD of 11% in this age interval (from 1.014 to 0.902 g/cm²). However, the parish population females displayed a greater decrease in BMD from middle to old adulthood than the high-status females (from 0.902 to 0.815 g/cm², - 8.6%). Interestingly, the parish females displayed a similar decrease in BMD early and late in life, i.e., from young to middle adulthood and middle to late adulthood. The decrease in BMD by middle adulthood in medieval females is far greater

**Table 5. Femur neck mean BMD values and SD for each sex, age group, and SES group.**

| | | | High status | | | | Parish pop. | | | Modern* | |
|---|---|---|---|---|---|---|---|---|---|---|---|
| | | n | BMD g/cm² | SD | %** | n | BMD g/cm² | SD | %** | BMD g/cm² | %** |
| FEMALES | YA | 12 | 1.094 | 0.09 | 100 | 14 | 1.014 | 0.16 | 100 | 0.985 | 100 |
| | MA | 9 | 0.898 | 0.15 | 82.1 | 22 | 0.902 | 0.22 | 89.0 | 0.943 | 96 |
| | OA | 4 | 0.975 | 0.07 | 89.1 | 40 | 0.815 | 0.15 | 80.4 | 0.843 | 86 |
| MALES | YA | 17 | 1.185 | 0.19 | 100 | 32 | 1.080 | 0.14 | 100.0 | 1.080 | 100 |
| | MA | 18 | 1.046 | 0.13 | 88.3 | 20 | 1.033 | 0.15 | 95.6 | 1.020 | 94 |
| | OA | 15 | 0.998 | 0.15 | 84.2 | 24 | 0.935 | 0.15 | 86.6 | 0.940 | 87 |

*Reference values for USA/Northern Europe [51].

** The age-related changes are given as percent of the mean BMD in the Young Adult age category.

Mean BMD values in the modern reference population are shown on the far right. YA: Young Adults (20–35 years), MA: Middle Adults (35–50 years) and OA: Old Adults (> 50 years).

than the reduction observed in modern females (- 4%). In addition, the decline in BMD by old adulthood is somewhat greater for the parish population females than for modern females (- 14%). In contrast, high-status females display a slightly smaller reduction by old adulthood.

The age-related reduction in BMD for high-status males was characterized by considerable bone loss from young to middle adulthood (from 1.185 to 1.046 g/cm$^2$, - 11.7%), by far exceeding the bone loss observed in parish population males (from 1.080 to 1.033 g/cm$^2$, - 4.4%). Interestingly, the early bone loss in the latter group was similar to that observed in modern men from young to middle adulthood (- 6%) [23, 75]. Mean BMD for males in the high-status group decreased further from middle to old adulthood (from 1.046 to 0.998 g/cm$^2$, - 4.1%). However, the parish population males experienced an even greater reduction in this age interval (from 1.033 to 0.935 g/cm$^2$, - 9%). The decrease in BMD by old adulthood was thus somewhat similar in these two status groups and comparable to the bone loss observed in modern men (0.940 g/cm$^2$, - 13%).

## Discussion

### SES differences reflected in BMD and stature

The extensive social stratification in the medieval period in Norway was reflected in BMD variation as detected in measurements of skeletal remains from five representative burials sites. Individuals from the parish population showed significantly lower BMD than individuals of high status, affecting both males and females. Non-similar SES did also affected stature: a low SES was associated with reduced height. Thus, individuals of high status presented a significantly higher BMD, and taller stature than individuals from the parish population did.

Several factors may have confounded the inferences made in this study regarding social stratification reflected in BMD and stature. For example, sampling bias due to preservation or the nature of the skeletal sample excavated from each burial site, small sample sizes due to strict inclusion criteria, methodological uncertainty, and subjectivity related to sex and age-at-death estimates.

### Living conditions during childhood and puberty

We did not detect a significant relationship between young adult BMD and SES for either females or males. These findings do not support the notion of distinct environmental factors significantly affecting young adult BMD in our two status groups, possibly indicating favorable conditions during childhood and puberty for both groups, allowing them to reach their skeletal potential.

It has been debated whether individuals of lower social status had access to a varied and adequate diet or if they suffered from mal- and/or undernourishment. The prevailing diet for the general population of Norway in this period has been described as monotonous and simple but wholesome and healthy (a "farmers' diet"), consisting of meat, fish, dairy products such as milk, butter, and cheese, porridge and bread [76]. Hufthammer [77] found few differences when comparing the diet of the medieval inhabitants of the Archbishop's Palace in Trondheim and shoemakers in Oslo. However, the upper-class individuals consumed a larger amount of game meat. Additionally, stature comparisons and prevalence of dental enamel hypoplasia in four cemeteries from medieval Norway (Bergen, Trondheim, Tønsberg, and Hamar) indicated that all social strata had access to a sufficient amount of food [78]. Kjellström, Storå Possnert and Linderholm [79] disclosed a dietary pattern reflecting the social hierarchy in the medieval town of Sigtuna, Sweden: individuals of high status consumed more animal protein than lower status individuals did. Yoder [80] detected a status-based difference in diet (types and quantity of food resources) between peasants, elites, and monks in a medieval Cistercian monastery in

Denmark. Research on medieval skeletal material from Northern Italy [81] demonstrated similar diets in childhood between high- and low-status individuals, followed by a shift to sex- and status-based differences in adult life. Overall, previous research on skeletal material from Norway and Scandinavia suggests that the general population of medieval Norway, regardless of SES, may have had a sufficient and varied diet and achieved their skeletal potential as indicated by peak BMD.

Holck and Kvaal [54] detected little evidence of enamel hypoplasia in the teeth of parishioners from the Church of St. Clemens, which indicated that these were not subject to malnourishment and disease (at least not to such a degree that it was reflected in their dentition). The remote and isolated location of Heidal (Church of Prestgard) indicated a more seasonal-based diet than at the other sites included in this study and a vulnerability in periods of crop failure [58]. This skeletal material was characterized by severe dental attrition, caries, plaque, tooth loss, and generally poor dental health, which was seen in connection with a diet mainly based on grains and flour containing grit (Holck P. Personal communication, 20.08. 2020). The consumption of porridge has a longstanding tradition in this area, and the locals probably ate varieties of this dish several times a day [82]. Sellevold [61] detected relatively few pathologies in the skeletal material from Hamar Cathedral, dental conditions and cribra orbitalia included, leading to the interpretation that this population had enjoyed beneficial living conditions during their years of growth and development.

Kersh, Martelli, Zebaze, Seeman and Pandy [83] stated that physical activity is perhaps the single most important lifestyle factor influencing peak bone mass. The benefits of physical activity/weight-bearing exercise and mechanical loading on BMD during childhood, adolescence, and young adulthood are well documented. Child labor was widespread in the medieval peasant community [84], which entailed an active daily lifestyle denoted by walking and weight-bearing exercise [85]. Children of the nobility learnt the art of combat from an early age, which involved daily training and hours of horseback riding [86]. However, their lifestyle was generally more sedentary and characterized by prolonged sitting [85]. This SES-related difference in physical activity may have positively impacted peak BMD in our parish population group compared to our high-status group.

Altogether, these findings do not indicate that the diet of the parish population juveniles in our sample was of low nutritional value, at least not to the degree that it resulted in distinct signs of malnourishment. In conclusion, it is likely that both our SES groups experienced favorable conditions during childhood and puberty, allowing them to reach their potential regarding peak BMD. However, this result could reflect the strong genetic control of peak BMD or the result of small or non-representative subsamples not reaching statistical significance.

Stature is among the most heritable traits in humans, but differences in mean stature between SES groups have indicated a possible environmental effect [34]. The observation that the mean stature for males in the high-status and parish populations was quite similar and concordant with the young adult BMD measurements indicates comparable living conditions during childhood and puberty. Admittedly, a diet characterized by extensive consumption of grain products (carbohydrates), such as that outlined for the Church of Prestgard, could result in lower stature than a diet rich in meat and fish (proteins) [87]. Since high young adult BMD and stature varied concordantly for males in both SES groups, we postulate that both these male populations likely had access to a varied and nutritious diet. Their living conditions during growth may have allowed them to reach their genetic height potential. The observation that females of high status were significantly taller than females in the parish population group suggests a possible impact of environment. It may indicate that these females experienced different conditions during growth and puberty. Overall, the analysis of stature in our sample supports the notion of distinct environmental factors in these two groups of females.

## Patterns of BMD variation across the adult life span

**Females: Greater SES differences?.**   The females in the two SES groups portray different trajectories of age-related reduction in BMD. The high-status females displayed distinct early bone loss followed by an increase in BMD, while the parish females displayed a non-distinct and similar reduction in BMD early and late in life.

Marked bone loss from young to middle adulthood has been linked to the nutritional strain of childbirth and lactation, so-called reproductive stress [4, 13, 15, 88]. However, a strict reproductive interpretation should be avoided [89]. Such pre- or peri-menopausal bone loss, as evident in the high-status females, has been observed in several previous studies of archaeological populations [13, 14, 24, 90, 91]. Compared to the modern population, the onset was earlier than today, the parity was high, and lactation was prolonged [4, 13, 14, 88–91]. Research on the effect of gestation on the maternal skeleton has given conflicting results, but most epidemiological studies documented a transient decrease in BMD connected to pregnancy and lactation [92, 93]. According to Stride, Patel and Kingston [94], first pregnancy in adolescence and a shorter reproductive lifespan between menarche and menopause, are associated with reduced BMD.

The marked early bone loss in the high-status group is indicates that these females experienced a greater depletion of bodily resources in this phase of life (e.g., early and repeated pregnancies). A sedentary lifestyle with considerable less physical activity among the high-status females may have contributed to a delayed recovery after pregnancy. The parish population females had probably been exposed to demanding physical labor connected to everyday activities, thus reducing the bone loss accompanying pregnancy and generally counteracting the struggles of everyday life.

Lees, Molleson, Arnett and Stevenson [20] linked the non-significant premenopausal bone loss in parish females buried at Christ Church, Spitalfields (18-19th century London) to the level of physical activity, both at work (weaving industry) and outside (walking), coupled with the bone conserving effects of parity. Recent research [95] has shown that the positive impact of parity on BMD is site-specific and the association with the femoral neck non-significant. The pattern of late rather than an early bone loss in women, so-called postmenopausal bone loss, has been observed in several previous studies of archaeological skeletal populations and is not a recent trait. It has been detected in skeletal material from the Early Bronze Age (4000 BP) in Austria [96], in a 3rd - 4th century CE population from Ancaster, UK [88], and in a Merovingian population (5th– 7th century CE) from Bockenheim, Germany [16]. We [23] previously observed a similar pattern in a post-Reformation skeletal material from Tangen Church, Norway, which was hypothesized to possibly indicate a different practice regarding childbirth and lactation in this aristocratic population, perhaps coupled with other as yet unidentified societal factors.

The increased BMD in elderly females in the high-status group may reflect nutritional and lifestyle factors influencing BMD from an early age. High-status females had such favorable living conditions enabling them to, e.g., recover from years of multiple pregnancies and depletion of bodily resources to a much larger degree than the parish population females. They could pay for upkeep and care in old age and would probably have enjoyed a varied and nutritious diet. When comparing mean BMD between SES groups for a specific sex and age group, we could only detect a significant difference between old adult females: the high status females had a significantly higher mean BMD than the parish females. The results are uncertain since the number of females in our high-status category was markedly smaller, with only four versus forty females from the parish population. However, the observed differences in BMD of both young and elderly females could be directly associated with socioeconomic conditions experienced by these groups.

Overall, we hypothesize that the different trajectories for females in our two SES groups may reflect greater socioeconomic differences for females than for males in the medieval society of Norway. This observation concurs with the concept of a strict sex division during this period. It is reasonable to assume that differences may be related to socio-cultural practices. Daily activities such as physical labor, habitual activities, dietary practices, housing, and childcare, likely affected BMD differently in our two SES groups. Whether medieval females were discriminated against in terms of nutrition has been debated. According to Benedictow [97], little information supports this hypothesis, at least when the Scandinavian countries are concerned. Stable isotope analyses performed by Kjellström, Storå, Possnert and Linderholm [79] disclosed a difference in dietary patterns between the sexes: females had a more homogenous diet than males, possibly explained by the fact that females were more stationary than males in medieval society.

**Males: More equal and privileged?.** The patterns of age-related bone loss observed in males in the two SES groups differed. The parish population group appeared to follow a modern trajectory since the age-related bone loss for males in this group followed the same pattern observed in modern men. Early bone loss in these males was similar to the bone loss observed in modern men in the same age interval [75]. The degree of late bone loss was also similar to that observed in modern men by old adulthood. We also observed this pattern of age-related bone loss (i.e., scarce early and significant late bone loss) in a previous study of a high-status population from the post-Reformation period in Norway [23]. We hypothesized that this pattern indicates that multiple factors, some of which are unidentified, influenced bone loss in this group. Mays, Lees and Stevenson [19] detected a significant late bone loss for males from medieval Wharram Percy, UK. As this bone loss was similar to or even exceeded the bone loss observed in modern males, the authors concluded that lifestyle factors might be less important than previously assumed.

The marked early bone loss observed in the high-status males is surprising and rarely observed in the archaeological literature. Interestingly, the early bone loss in these high-status males is similar to that observed in parish females. We interpret this as an indication of distinct environmental stresses or factors in the two groups during this life phase. The high social status of the males in this group probably entailed good housing conditions, access to a varied and wholesome diet, and favorable sanitary conditions. On the other hand, their status probably led to a more sedentary life, which could accelerate bone loss. Combined with lifestyle risk factors predisposing to "modern diseases" such as metabolic syndrome and diabetes 2, progressive bone deterioration would develop. Research on alcoholism in medieval England states that excessive consumption of alcohol was widely spread in all classes of society but prevailed among the clergy and university students [98]. A previous study [23] detected significant early bone loss (p = 0.03, q = 0.09, two sample t-test, Benjamini-Hochberg procedure) in males in a skeletal material dated to the Late Iron Age in Norway. This early bone loss was followed by a slight increase in BMD from middle to old adulthood. Many of these burials were considered characteristic of the upper social strata at the time. Strenuous physical activity, poor (childhood) nutrition, shorter life expectancy, and demanding living conditions were all considered explanatory factors.

The degree of late bone loss in males in the high-status group was markedly less than that observed in parish males and modern men, possibly indicating more favorable conditions for elderly males of high social status than for elderly males in the parish population. For example, their position could entail better housing and dietary practices, as well as the ability to pay for provent and (medical) care in old age. Interestingly, the overall reduction in BMD by old adulthood appeared somewhat similar for males in our two status groups and comparable to the bone loss observed in modern men by old adulthood.

## Conclusion

We examined BMD in skeletal remains from five burial sites reflecting medieval Norway and detected a significant difference in BMD related to SES. Individuals from the parish population had a significantly lower BMD than individuals of high status. Moreover, greater socioeconomic and sociocultural differences for females than males were observed in the medieval society of Norway, demonstrating the impact of living conditions and nutrition on growth and skeletal development as reflected in BMD variation and stature. Our findings indicate that femur neck BMD may be a valuable skeletal indicator of SES.

## Supporting information

**S1 Table. Data on age, sex, SES, BMD and stature for the total study sample.**
(PDF)

## Acknowledgments

We want to acknowledge the contribution of the following: Principal supervisor Trygve Brauns Leergaard for article revisions, practical advice, and thesis management. The contemporary Dominican Congregation at St. Dominicus Convent in Oslo, for correspondence and information on archive literature. We thank Kari Ormstad at the Department of Forensic Sciences, Oslo University Hospital, for insightful comments and constructive suggestions. We would also like to thank Carina V. S. Knudsen at the Section of Internal Service, IMB, UiO, for diligent work with article illustrations.

## Author Contributions

**Conceptualization:** Elin T. Brødholt, Kaare M. Gautvik.

**Data curation:** Elin T. Brødholt.

**Formal analysis:** Elin T. Brødholt, Clara-Cecilie Günther.

**Investigation:** Elin T. Brødholt.

**Methodology:** Elin T. Brødholt, Kaare M. Gautvik.

**Project administration:** Elin T. Brødholt.

**Resources:** Elin T. Brødholt.

**Visualization:** Elin T. Brødholt, Clara-Cecilie Günther.

**Writing – original draft:** Elin T. Brødholt, Kaare M. Gautvik, Clara-Cecilie Günther.

**Writing – review & editing:** Elin T. Brødholt, Kaare M. Gautvik, Clara-Cecilie Günther, Torstein Sjøvold, Per Holck.

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
