## [Decision Letter · Decision Letter 0]

4 Jul 2022

PONE-D-21-33496Social stratification reflected in bone mineral density: Spectral imaging and osteoarchaeological findings from medieval NorwayPLOS ONE

Dear Dr. Brødholt,

Thank you for submitting your manuscript to PLOS ONE. After careful consideration, we feel that it has merit but does not fully meet PLOS ONE’s publication criteria as it currently stands. Therefore, we invite you to submit a revised version of the manuscript that addresses the points raised during the review process. Both reviewers provide suggestions for clarifying and improving your manuscript. Please address each of the comments and suggestions while making your revisions.

We look forward to receiving your revised manuscript.

Kind regards,

John P. Hart, Ph.D.

Academic Editor

PLOS ONE

Journal Requirements:

2. In your manuscript, please provide additional information regarding the specimens used in your study. Ensure that you have reported specimen numbers and complete repository information, including museum name and geographic location.

For more information on PLOS ONE's requirements for paleontology and archaeology research, see https://journals.plos.org/plosone/s/submission-guidelines#loc-paleontology-and-archaeology-research.

4. We note that Figures 1, 2, 3 and 4 in your submission contain map images which may be copyrighted. All PLOS content is published under the Creative Commons Attribution License (CC BY 4.0), which means that the manuscript, images, and Supporting Information files will be freely available online, and any third party is permitted to access, download, copy, distribute, and use these materials in any way, even commercially, with proper attribution. For these reasons, we cannot publish previously copyrighted maps or satellite images created using proprietary data, such as Google software (Google Maps, Street View, and Earth). For more information, see our copyright guidelines: http://journals.plos.org/plosone/s/licenses-and-copyright.

a. You may seek permission from the original copyright holder of Figures 1, 2, 3 and 4 to publish the content specifically under the CC BY 4.0 license.  

Reviewers' comments:

Reviewer's Responses to Questions

**Comments to the Author**

1. Is the manuscript technically sound, and do the data support the conclusions?

Reviewer #1: Yes

Reviewer #2: Yes

2. Has the statistical analysis been performed appropriately and rigorously? 

Reviewer #1: Yes

Reviewer #2: Yes

3. Have the authors made all data underlying the findings in their manuscript fully available?

Reviewer #1: Yes

Reviewer #2: Yes

4. Is the manuscript presented in an intelligible fashion and written in standard English?

Reviewer #1: Yes

Reviewer #2: Yes

5. Review Comments to the Author

Reviewer #1: This study examined BMD obtained via DEXA to test for differences between socio-economic status (SES) groups in Medieval Norway. Associations with stature are also considered. The finding that higher SES individuals are statistically significantly taller and have higher BMD than those of lower SES makes sense in the light of bone biology literature. I think this is a very thorough study that uses a large sample size and provides further data for the detrimental effects of social disadvantage on bone health. I have noticed some shortcomings here and there that I outline constructively for the authors in the hope that revisions will strengthen the manuscript.

Introduction: I suggest the authors unpack the association between BMD and stature given that stature is included in the analyses as a variable. Further, the Introduction would benefit from an additional paragraph discussing specifically what has already been found in the archaeological literature in terms of BMD/bone health measures and status aside from Norway. The readership of PLOS One is quite broad, so I don’t think we can expect that readers will be well versed in the bioarchaeology literature.

Methods: I think the authors should provide a justification for the statistical procedures – e.g. explain how assumptions about data were validated and what/whether normality tests were used to justify the selection of inferential tests.

Results: When describing the stature results, I would encourage the authors to also cite SD results in parentheses (I know these are shown in the tables but I’m sure the readers would appreciate the SD data in cm in text). That is because these are estimates rather than raw measurements, so they’re not entirely accurate.

Discussion: I would just encourage the authors to add a paragraph explicitly stating the limitations of this study. The biggest elephant in the room is probably the fact that the sex and age-at-death estimates are probability scores, essentially. So, the authors are having to rely on these estimates for their sex and age-at-death comparisons. Of course, this is something we cannot overcome in bioarchaeology, so acknowledging such issues is the way to go, I think.

Writing mechanics:

- Abstract: this sentence is not clear “We detected a significant association between bone mineral density and socioeconomic status on one side and stature variations on the other.” – I’m not sure what the authors mean by ‘side’.

- Throughout, ‘age’ should probably be ‘age-at-death’.

- I’m not sure the authors should be using the word ‘gender’ – there’s been a lot of discussion in the bioarchaeological literature recently about the need to differentiate between gender and sex.

- I also think the authors shouldn’t be pluralising ‘variation’.

Thank you for the opportunity to review this interesting manuscript.

Justyna Miszkiewicz, PhD

Australian National University

Reviewer #2: This manuscript is a successor to Brødholt et al. 2021 written by the same authors, based on archaeological skeletal material from Norway, covering the time-span from Late Iron Age to post-Reformation period. The current study focuses solely on medieval populations. The bone mineral density data obtained in the previous study and estimated stature were analysed in relation to socio-economic status. Results of present research indicate that BMD measured in the femoral neck area may be a valuable skeletal indicator of SES.

This is a interesting study, because similar research is limited in number, however, extensive revision is necessary before the study is suitable for publication.

First of all, in my opinion the title does not fully reflect the content of the work. There is no mention of stature, which was extensively discussed throughout the manuscript.

Abstract - The aim of the abstract is to convey essential information and familiarize the reader with the most important research findings without having to go deeper into the text. Therefore, please rewrite the abstract and supplement the basic information about the total number of males and females (we only know that the research material consists of 227 individuals) and information about the division of the study group into age categories. When describing the most important research results, please provide tests' results or statistical significance. Information on the differences in BMD in women is missing in line 25. Authors should also indicate what environmental factors were analysed (line 28).

Introduction - line 52 - please add at least 2-3 references in square brackets to illustrate "numerous studies"

- line 55 – please expand the statement "variety of age-related patterns"

- line 58-59 - providing general information about the existence of differences in BMD values among archaeological Scandinavian populations from different historical periods does not seem to be sufficient. Please briefly describe these differences and indicate the presence or absence of a trend

- line 60 contains the unfortunate phrase "living skeleton" - please rephrase this sentence

- line 60-62 - the sentence is too general, please explain briefly the relationship between SES and BMD in contemporary human populations and in the studies carried out so far on historical human populations

- line 61 - to use an abbreviation for the first time, it needs to be expanded first. The expansion of DXA abbreviation does not appear until line 73

- line 66-67 - the sentence suggests that the assignment of the 5 archaeological sites to the selected SES category was made in present study, however, the SES of 4 out of 5 population was already identified in the authors' previous article. Only for St. Olav Monastery the SES was clarified in the present study. This should be included in the description by adding the appropriate citation.

- There is no reference to the stature in the aim of the present research, although this parameter is also measured and analysed in relation to SES.

- line 71 - the phrase "medieval period" seems to be unnecessary. In the same sentence it is mentioned earlier that the research material comes from medieval burial sites

Materials - The descriptions of archaeological burial sites are important, however, I propose to shorten them and leave the most important information indicating the socio-economic status of the buried people. For example, the information in lines 84, 94, 113, 125, 144 does not seem to be relevant to the issue under discussion.

Methods - line 165 - information about DXA analysis in Osteological analysis paragraph seems to be unnecessary

line 165-166 - the authors report that the assessment of sex, age, stature, pathology and trauma was carried out using the methods proposed by Buikstra and Ubelaker (1994). However, in the further part of the paragraph, they state that they used methods proposed by other researchers to assess, for example: sex, age or stature. Buikstra and Ubelaker (1994) are only mentioned in the context of distinguishing age categories. Please systematize the methods in the Osteological analysis paragraph

- line 180 – latin names should be written in italics

- line 185 - please provide the abbreviation of the procedure in brackets after its full name

- line 188-189 - The authors inform that 71 individuals with BMD measurements were included in the study (instead of 227). Probably the number of individuals was given here by mistake, please correct it.

Results – line 202 – please change “gender” to “sex” which is more appropriate term when analysing human historical populations (also in other part of manuscript)

- line 203-206 – please use digits instead of verbally notation when providing number of individuals

- line 207 - Figure 5 seems to be superfluous. It does not contain any information that is not also included in Table 2.

- line 215-216 - please clarify the sentence, as it currently contradicts the information from line 209

- line 268 - please replace Brødholt, Günther, Gautvik, Sjøvold and Holck (15) with Brødholt et al. [15]- the same situation in line 309, 335, 385, 416, 429

- line 290 – reference [67] seems to be unjustified in this paragraph

Discussion - Living conditions during childhood and puberty paragraph - The information contained in the this part of Discussion section is substantively correct, however, the manner of its presentation does not build a coherent whole aimed at relating the results obtained in the current research to the existing knowledge on archaeological burial sites and SES.

- line 333 - contains an incorrectly formed phrase "parish population subadultus" - please rephrase this sentence

- line 439 - the authors suggest that a sedentary lifestyle can lead to inactivation of bone loss, while literature data provides us with information that low physical activity and a sedentary lifestyle promote bone loss leading to osteoporosis. Perhaps the word inactivation was used incorrectly and should be "acceleration" or “activation”

- Table 1 - according to the caption, Table 1 should contain data of individuals included in the research. Instead, it also includes 20 individuals that were excluded from the study. The total number of individuals is therefore 247 (bolded) instead of 227. This is confusing and inconsistent with the table header. Please correct data in Table 1.

Additionally, the data for Table 1 were taken from the authors' earlier work (Brødholt et al. 2021). In order not to let the authors be accused of self-plagiarism, please include adequate citation in the caption of the table with the annotation "modified".

- S1 Table - please limit the BMD values to 3 decimal places

General:

- Too long sentences make the reception of the manuscript difficult. Please use shorter sentences.

- Please review the text again carefully for typos and minor linguistic errors

- Please cite all reference number in square brackets – e.g. lines 188, 268 etc., also in figures’ captions

- Please unify the AD / A.D. notation. The correct version is AD

6. PLOS authors have the option to publish the peer review history of their article (what does this mean?). If published, this will include your full peer review and any attached files.

Reviewer #1: **Yes: **Justyna Miszkiewicz

Reviewer #2: No

---

## [Author Response · Author response to Decision Letter 0]

18 Aug 2022

RESPONSE TO EDITOR AND REVIEWER COMMENTS

We thank the editor and the two reviewers for their valuable and constructive comments and suggestions, and we hope they will find these revisions agreeable. All suggested changes are accommodated, and we have provided additional information when available.

The following is a point-by-point response to the questions and comments in your letter dated July 4, 2021. We refer to page and line numbers in the revised document.

Sincerely,

On behalf of all authors,

Elin T. Brødholt

Department of Molecular Medicine, Institute of Basic Medical Sciences, Faculty of Medicine, University of Oslo

0372 Oslo, Norway

+47 47878202 

e.t.brodholt@medisin.uio.no

ACADEMIC EDITOR:

• RESPONSE: 

Thank you, we have edited the manuscript for it to comply with the style requirements. Hopefully, you will find these revisions agreeable. 

2. In your manuscript, please provide additional information regarding the specimens used in your study. Ensure that you have reported specimen numbers and complete repository information, including museum name and geographic location.

For more information on PLOS ONE's requirements for paleontology and archaeology research, see https://journals.plos.org/plosone/s/submission-guidelines#loc-paleontology-and-archaeology-research.

• RESPONSE: 

We have provided information on the specimens used in our study in the Materials section of the manuscript (page 4, lines 94-96): “The skeletal material from the five medieval burial sites included in our study (Fig 1) spans from the 11th to the 16th century AD in Norway. It constitutes part of the Schreiner Collection at the Division of Anatomy, University of Oslo.”

Additionally, we have edited our Supplementary data (S1 Table) to include complete repository information and the geographical location of the burial sites. Information on specimen numbers is also given.

We have also provided details for all permits obtained, including the full name of the issuing authority. In addition, we have added the following statement regarding necessary permits (page 6, lines 151-153): “Ethical approval was obtained from the National Committee for Research Ethics on Human Remains, The Norwegian National Research Ethics Committees (2015/396 and 2016/304). All necessary permits were obtained for the described study, which complied with all relevant regulations.” 

• RESPONSE: 

The study’s minimal data set is given in the Supporting Information file (S1 Table), which we have revised.

We would be grateful if you could update our Data Availability Statement by adding the following sentence: “All relevant data are within the manuscript and its Supporting Information files.”

4. We note that Figures 1, 2, 3, and 4 in your submission contain map images which may be copyrighted. All PLOS content is published under the Creative Commons Attribution License (CC BY 4.0), which means that the manuscript, images, and Supporting Information files will be freely available online, and any third party is permitted to access, download, copy, distribute, and use these materials in any way, even commercially, with proper attribution. For these reasons, we cannot publish previously copyrighted maps or satellite images created using proprietary data, such as Google software (Google Maps, Street View, and Earth). For more information, see our copyright guidelines: http://journals.plos.org/plosone/s/licenses-and-copyright.

a. You may seek permission from the original copyright holder of Figures 1, 2, 3 and 4 to publish the content specifically under the CC BY 4.0 license. 

• RESPONSE: 

Thank you for your suggestions. We have removed the figures in question (Figures. 1-4) from our submission and replaced them with a new figure (Figure 1) based on map data from U.S. Geological Survey. In addition, we have added the following credit to our Figure 1 caption (page 5, line 102); “Figure created with map data from U.S. Geological Survey.”

We hope you will find our new figure agreeable.

REVIEWER 1 COMMENTS:

This study examined BMD obtained via DEXA to test for differences between socio-economic status (SES) groups in Medieval Norway. Associations with stature are also considered. The finding that higher SES individuals are statistically significantly taller and have higher BMD than those of lower SES makes sense in the light of bone biology literature. I think this is a very thorough study that uses a large sample size and provides further data for the detrimental effects of social disadvantage on bone health. I have noticed some shortcomings here and there that I outline constructively for the authors in the hope that revisions will strengthen the manuscript.

• RESPONSE: 

Thank you for your positive review and for your valuable comments and suggestions. All suggested changes have been accommodated, and we feel that the comments by the reviewers have much improved the paper. We hope you will find these revisions agreeable. 

Introduction: I suggest the authors unpack the association between BMD and stature given that stature is included in the analyses as a variable. Further, the Introduction would benefit from an additional paragraph discussing specifically what has already been found in the archaeological literature in terms of BMD/bone health measures and status aside from Norway. The readership of PLOS ONE is quite broad, so I don’t think we can expect that readers will be well versed in the bioarchaeology literature.

• RESPONSE:

Thank you for providing these insights and suggestions. We agree and have rewritten this paragraph and added information on the association between BMD and stature (page 4, lines 82-88); “Research has shown that stature reflects SES strongly and consistently, and this positive association is widely observed across cultures [26, 29]. Stature is among the most heritable traits in humans, but differences in stature between SES groups have shown that it is modified by environmental factors. These factors are particularly important during development [30-33]. Furthermore, research has revealed that the crucial periods for bone mineralization in non-adults overlap with those for stature attainment [26]. We, therefore, implemented stature as a variable in our analyses.”

We have also added a paragraph on results from previous research on archaeological populations regarding BMD and SES (as suggested by both reviewers) (page 3, lines 68-78); “Research on BMD variation, as measured by Dual Energy X-Ray Absorptiometry (DXA), related to SES in archaeological skeletal material has been limited [4-6]. However, the findings demonstrate the complex and not so straightforward relationship between SES, sex, and lifestyle factors such as nutrition and physical activity. For example, Di Stefano, Boano [6] and Borre, Boano [28] found that high-status individuals buried inside the San Michele Church, North-West Italy, had significantly lower BMD than individuals buried in the cemetery. A larger intake of dairy products, more sun exposure, and greater physical activity in the cemetery group were presumed to explain the observed pattern. Zaki, Hussien [4] investigated the occurrence of osteoporosis in two social classes from the Old Kingdom in Giza and found a higher prevalence in male workers compared to male high officials and female high officials versus female workers. The authors related the findings to nutritional stress and increased workload in male workers, and a sedentary lifestyle among female high officials.”

Methods: I think the authors should provide a justification for the statistical procedures – e.g. explain how assumptions about data were validated and what/whether normality tests were used to justify the selection of inferential tests.

• RESPONSE: 

You have raised an important question. The distributions of the response variables neck mean BMD and stature were visually inspected by histograms and QQ-plots and we concluded that the data were approximately normally distributed. The residual plots of the regression models showed that the model assumptions were met. We have added the following sentences to the Statistical analysis section (page 8, lines 181-183): " The distributions of neck mean BMD and stature were first visually inspected by histograms and QQ-plots and found to be approximately normally distributed. " and (page 8, lines 187-188): "Model assumptions of independence and normality were checked by using residual plots."

Results: When describing the stature results, I would encourage the authors to also cite SD results in parentheses (I know these are shown in the tables but I’m sure the readers would appreciate the SD data in cm in text). That is because these are estimates rather than raw measurements, so they’re not entirely accurate.

• RESPONSE:

We have rewritten this paragraph to include information on SD (page 8, lines 201-203); “The mean statures in the high-status group were 175.4 cm (SD 5.0) for males and 164.2 cm (SD 5.9) for females, while the mean statures in the parish group were 174.6 cm (SD 5.8) for males and 160.7 cm (SD 4.9) for females (Table 2).”

Discussion: I would just encourage the authors to add a paragraph explicitly stating the limitations of this study. The biggest elephant in the room is probably the fact that the sex and age-at-death estimates are probability scores, essentially. So, the authors are having to rely on these estimates for their sex and age-at-death comparisons. Of course, this is something we cannot overcome in bioarchaeology, so acknowledging such issues is the way to go, I think.

• RESPONSE: 

Thank you for spotting this missing information. We have added a paragraph on limitations of this study at the start of the discussion (page 13-14, lines 293-296); “Several factors may have confounded the inferences made in this study regarding social stratification reflected in BMD and stature. For example, sampling bias due to preservation or the nature of the skeletal sample excavated from each burial site, small sample sizes due to strict inclusion criteria, methodological uncertainty, and subjectivity related to sex and age-at-death estimates.”

Writing mechanics:

- Abstract: this sentence is not clear “We detected a significant association between bone mineral density and socioeconomic status on one side and stature variations on the other.” – I’m not sure what the authors mean by ‘side’.

- Throughout, ‘age’ should probably be ‘age-at-death’.

- I’m not sure the authors should be using the word ‘gender’ – there’s been a lot of discussion in the bioarchaeological literature recently about the need to differentiate between gender and sex.

- I also think the authors shouldn’t be pluralising ‘variation’.

• RESPONSE:

Thank you for your suggestions. We have corrected these grammatical issues and copy-edited the manuscript.

REVIEWER 2 COMMENTS:

This manuscript is a successor to Brødholt et al. 2021 written by the same authors, based on archaeological skeletal material from Norway, covering the time-span from Late Iron Age to post-Reformation period. The current study focuses solely on medieval populations. The bone mineral density data obtained in the previous study and estimated stature were analysed in relation to socio-economic status. Results of present research indicate that BMD measured in the femoral neck area may be a valuable skeletal indicator of SES.

This is a interesting study, because similar research is limited in number, however, extensive revision is necessary before the study is suitable for publication.

• RESPONSE: 

Thank you for reviewing our paper and for all your constructive comments and suggestions. All suggested changes have been accommodated, and we feel that the comments by the reviewers have much improved the paper. We hope you will find these revisions agreeable.

First of all, in my opinion the title does not fully reflect the content of the work. There is no mention of stature, which was extensively discussed throughout the manuscript.

• RESPONSE: 

We agree with you and have modified the title to include stature: “Social stratification reflected in bone mineral density and stature: Spectral imaging and osteoarchaeological findings from medieval Norway”.

Abstract - The aim of the abstract is to convey essential information and familiarize the reader with the most important research findings without having to go deeper into the text. Therefore, please rewrite the abstract and supplement the basic information about the total number of males and females (we only know that the research material consists of 227 individuals) and information about the division of the study group into age categories. When describing the most important research results, please provide tests' results or statistical significance. Information on the differences in BMD in women is missing in line 25. Authors should also indicate what environmental factors were analysed (line 28).

• RESPONSE:

Thank you for pointing out this missing information in our abstract. We have rewritten the abstract and included the suggested information (page 1, lines 8-23); “We combine osteological analysis and Dual Energy X-Ray Absorptiometry, studying the remains of 227 individuals (102 females and 125 males) employing young, middle, and old adult age categories. The aim is to assess bone mineral density as a skeletal indicator of socioeconomic status including stature as a variable. We detected that socioeconomic status significantly affected bone mineral density and stature. Individuals of high status had higher bone mineral density (0.07 g/cm², p = 0.003) and taller stature (1.85 cm, p = 0.017) than individuals from the parish population. We detected no significant relationship between young adult bone mineral density and socioeconomic status (p= 0.127 and 0.059 for females and males, respectively). For males, high young adult bone mineral density and stature varied concordantly in both status groups. In contrast, females of high status were significantly taller than females in the parish population (p = 0.011). Our findings indicate quite different conditions during growth and puberty for the two groups of females. The age-related pattern of bone variation also portrayed quite different trajectories for the two socioeconomic status groups of both sexes. We discuss sociocultural practices (living conditions during childhood and puberty, as well as nutritional and lifestyle factors in adult life), possibly explaining the differences in bone mineral density between the high-status and parish population groups.”

Introduction - line 52 - please add at least 2-3 references in square brackets to illustrate "numerous studies"

• RESPONSE: 

Thank you for spotting this missing information. We have rewritten this sentence (page 3, lines 54-56); “The high occurrence of osteoporosis in modern Caucasian populations led to numerous studies focusing on this condition in archaeological skeletal remains [11-20], searching to unveil if patterns of bone loss in the past resembled those observed in the modern population.”

- line 55 – please expand the statement "variety of age-related patterns"

• RESPONSE:

Thank you for your suggestion. We have rewritten this sentence (page 3, lines 57-59); “Instead, these studies demonstrated a varied pattern of age- and sex-related bone loss: ranging from less to generally similar or greater bone loss than in modern populations.”

- line 58-59 - providing general information about the existence of differences in BMD values among archaeological Scandinavian populations from different historical periods does not seem to be sufficient. Please briefly describe these differences and indicate the presence or absence of a trend

• RESPONSE:

Thank you for these insights. We have rewritten this paragraph (page 3, lines 62-65); “Bone mineral density (BMD) has varied notably between archaeological populations and time periods in Scandinavia [4, 12-15, 17, 23, 24]. The age- and sex-related BMD and patterns of bone loss in these populations are diverse, and the findings demonstrate the lack of consistent trends. It has been unclear to what extent this variation in BMD can be linked to social inequality.”

- line 60 contains the unfortunate phrase "living skeleton" - please rephrase this sentence

• RESPONSE: 

Thank you for your suggestion. We have rewritten this sentence (page 3, lines 65-70); “Clinical studies [25-27] demonstrate the effect of SES on skeletal BMD; however, they conclude that it is difficult to establish a consistent and conclusive positive or negative association. Education and income are positively associated with BMD but depend on factors such as sex, age, and ethnicity. Research on BMD variation, as measured by Dual Energy X-Ray Absorptiometry (DXA), related to SES in archaeological skeletal material has been limited [4-6].” 

- line 60-62 - the sentence is too general, please explain briefly the relationship between SES and BMD in contemporary human populations and in the studies carried out so far on historical human populations

• RESPONSE: 

Thank you for this constructive suggestion. We have rewritten this paragraph to state the current research status and clarify the complex relationship between SES and BMD (page 3-4, lines 65-78): “Clinical studies [25-27] demonstrate the effect of SES on skeletal BMD; however, they conclude that it is difficult to establish a consistent and conclusive positive or negative association. Education and income are positively associated with BMD but depend on factors such as sex, age, and ethnicity. Research on BMD variation, as measured by Dual Energy X-Ray Absorptiometry (DXA), related to SES in archaeological skeletal material has been limited [4-6]. However, the findings demonstrate the complex and not so straightforward relationship between SES, sex, and lifestyle factors such as nutrition and physical activity. For example, Di Stefano, Boano [6] and Borre, Boano [28] found that high-status individuals buried inside the San Michele Church, North-West Italy, had significantly lower BMD than individuals buried in the cemetery. A larger intake of dairy products, more sun exposure, and greater physical activity in the cemetery group were presumed to explain the observed pattern. Zaki, Hussien [4] investigated the occurrence of osteoporosis in two social classes from the Old Kingdom in Giza and found a higher prevalence in male workers compared to male high officials and female high officials versus female workers. The authors related the findings to nutritional stress and increased workload in male workers, and a sedentary lifestyle among female high officials.” 

- line 61 - to use an abbreviation for the first time, it needs to be expanded first. The expansion of DXA abbreviation does not appear until line 73

• RESPONSE: 

Thank you, we have corrected this issue.

- line 66-67 - the sentence suggests that the assignment of the 5 archaeological sites to the selected SES category was made in present study, however, the SES of 4 out of 5 population was already identified in the authors' previous article. Only for St. Olav Monastery the SES was clarified in the present study. This should be included in the description by adding the appropriate citation.

• RESPONSE: 

Thank you for spotting this missing information. We have rewritten this sentence (page 4, lines 88-89): “Based on previous research by the authors [23], we have classified the five burial sites included in the study as essentially reflecting two SES groups.”

- There is no reference to the stature in the aim of the present research, although this parameter is also measured and analysed in relation to SES.

RESPONSE: 

Thank you for spotting this missing information. We have rewritten this paragraph to explain the inclusion of stature as a variable in our analysis (page 4, lines 82-88); “Research has shown that stature reflects SES strongly and consistently, and this positive association is widely observed across cultures [26, 29]. Stature is among the most heritable traits in humans, but differences in stature between SES groups have shown that it is modified by environmental factors. These factors are particularly important during development [30-33]. Furthermore, research has revealed that the crucial periods for bone mineralization in non-adults overlap with those for stature attainment [26]. We, therefore, implemented stature as a variable in our analyses.” 

- line 71 - the phrase "medieval period" seems to be unnecessary. In the same sentence it is mentioned earlier that the research material comes from medieval burial sites

• RESPONSE: 

Thank you, we have corrected this issue.

Materials - The descriptions of archaeological burial sites are important, however, I propose to shorten them and leave the most important information indicating the socio-economic status of the buried people. For example, the information in lines 84, 94, 113, 125, 144 does not seem to be relevant to the issue under discussion.

• RESPONSE: 

Thank you for your suggestion. We have shortened the description of the included burial sites (page 5-6, lines 104-148). Please see the revised manuscript for details.

Methods - line 165 - information about DXA analysis in Osteological analysis paragraph seems to be unnecessary

• RESPONSE: 

Thank you for your suggestion. We have rewritten this sentence (page 7, lines 157-158); “All skeletal remains were subject to a separate evaluation of sex, age at death, stature, pathology, and trauma.”

line 165-166 - the authors report that the assessment of sex, age, stature, pathology and trauma was carried out using the methods proposed by Buikstra and Ubelaker (1994). However, in the further part of the paragraph, they state that they used methods proposed by other researchers to assess, for example: sex, age or stature. Buikstra and Ubelaker (1994) are only mentioned in the context of distinguishing age categories. Please systematize the methods in the Osteological analysis paragraph

• RESPONSE: 

Thank you for pointing out this issue. We have rewritten this paragraph to clarify which methods are proposed by Buikstra and Ubelaker [47] (page 7, lines 158-164); “We conducted this evaluation according to traditional methods given by Buikstra and Ubelaker [61]; sexing of crania [62], assessment of pelvic features [63], estimation of age-at-death was performed by evaluating suture closure [64] as well as the pubic symphysis [65, 66]. We estimated stature according to Trotter and Gleser [67], Trotter and Gleser [68], and documented pathology and trauma following Ortner [69] and Aufderheide and Rodríguez-Martín [70]. We applied broad age groups: Young Adult (20-35 years), Middle Adult (35-50 years), and Old Adult (50+), as per Buikstra and Ubelaker [61].”

- line 180 – latin names should be written in italics

• RESPONSE: 

Thank you, we have corrected this issue.

- line 185 - please provide the abbreviation of the procedure in brackets after its full name

• RESPONSE: 

Thank you, we have corrected this issue

- line 188-189 - The authors inform that 71 individuals with BMD measurements were included in the study (instead of 227). Probably the number of individuals was given here by mistake, please correct it.

• RESPONSE: 

Thank you for spotting this issue. We have rewritten this section to clarify that we refer to the 71 individuals previously examined by Holck (page 7, lines 176-178): “The study adopted BMD data from the DXA analysis by Holck [4], and we included 71 individuals with scans in accordance with the DXA analysis criteria.”

Results – line 202 – please change “gender” to “sex” which is more appropriate term when analysing human historical populations (also in other part of manuscript)

• RESPONSE: 

Thank you for your suggestion. We have corrected this issue and used the term “sex” throughout the manuscript.

- line 203-206 – please use digits instead of verbally notation when providing number of individuals

• RESPONSE: 

Thank you for your suggestion. We have corrected this issue throughout the manuscript.

- line 207 - Figure 5 seems to be superfluous. It does not contain any information that is not also included in Table 2.

• RESPONSE: 

We agree and have removed this figure.

- line 215-216 - please clarify the sentence, as it currently contradicts the information from line 209

• RESPONSE:

The regression model is fitted to the full dataset, i.e. all observations for both sexes and socioeconomic groups. We find that socioeconomic status has a significant effect on stature. The t-tests are performed on smaller subsets, for females and males separately, and the effect of socioeconomic status is not significant for both groups in this subsets.

We have added the following clarification in our manuscript (page 8-9, lines 203-205); “When considering the data for each sex separately, females in the high-status group were significantly taller than females in the parish population group (3.5 cm, p = 0.011, two sample t-test, Table 4). No statistical difference was detected for males (0.8 cm, p = 0.410).”

- line 268 - please replace Brødholt, Günther, Gautvik, Sjøvold and Holck (15) with Brødholt et al. [15]- the same situation in line 309, 335, 385, 416, 429

• RESPONSE: 

Thank you for spotting this formatting issue. We have corrected this throughout the manuscript.

- line 290 – reference [67] seems to be unjustified in this paragraph

• RESPONSE:

The information regarding bone loss observed in modern men is obtained from reference [67] (DXA User Manual) and these data were recalculated (percentage loss) in our previous publication [15]. 

Discussion - Living conditions during childhood and puberty paragraph - The information contained in the this part of Discussion section is substantively correct, however, the manner of its presentation does not build a coherent whole aimed at relating the results obtained in the current research to the existing knowledge on archaeological burial sites and SES.

• RESPONSE: 

Thank you for these insights. We fully agree, and have rewritten this entire paragraph for it to be more coherent. We have attempted to connect our results to existing knowledge (page 14-16, lines 298-356). Please see the revised manuscript for details.

- line 333 - contains an incorrectly formed phrase "parish population subadultus" - please rephrase this sentence

• RESPONSE: 

We have rewritten this sentence (page 15, lines 338-340); “Altogether, these findings do not indicate that the diet of the parish population juveniles in our sample was of low nutritional value, at least not to the degree that it resulted in distinct signs of malnourishment.”

- line 439 - the authors suggest that a sedentary lifestyle can lead to inactivation of bone loss, while literature data provides us with information that low physical activity and a sedentary lifestyle promote bone loss leading to osteoporosis. Perhaps the word inactivation was used incorrectly and should be "acceleration" or “activation”

• RESPONSE: 

Thank you for spotting this mistake. We have rewritten this sentence (page 18-19, lines 426-427); “On the other hand, their status probably led to a more sedentary life, which could accelerate bone loss.”

- Table 1 - according to the caption, Table 1 should contain data of individuals included in the research. Instead, it also includes 20 individuals that were excluded from the study. The total number of individuals is therefore 247 (bolded) instead of 227. This is confusing and inconsistent with the table header. Please correct data in Table 1.

Additionally, the data for Table 1 were taken from the authors' earlier work (Brødholt et al. 2021). In order not to let the authors be accused of self-plagiarism, please include adequate citation in the caption of the table with the annotation "modified".

• RESPONSE:

We agree with you and have corrected the data in Table 1 (page 9). We have also added a reference to our previous work in the caption of Table 1 (page 8, line 206): “Table modified from Brødholt, Günther [13]”.

- S1 Table - please limit the BMD values to 3 decimal places

• RESPONSE: 

Thank you, we have corrected this issue in our new S1 Table.

General:

- Too long sentences make the reception of the manuscript difficult. Please use shorter sentences.

- Please review the text again carefully for typos and minor linguistic errors

- Please cite all reference number in square brackets – e.g. lines 188, 268 etc., also in figures’ captions

- Please unify the AD / A.D. notation. The correct version is AD

• RESPONSE: 

Thank you for providing these suggestions. We have corrected these issues, and the manuscript has been copy-edited. 

Sincerely,

Elin T. Brødholt

Department of Molecular Medicine, University of Oslo

0372 Oslo, Norway

+47 47878202

e.t.brodholt@medisin.uio.no

---

## [Decision Letter · Decision Letter 1]

19 Sep 2022

Social stratification reflected in bone mineral density and stature: Spectral imaging and osteoarchaeological findings from medieval Norway

PONE-D-21-33496R1

Dear Dr. Brødholt,

We’re pleased to inform you that your manuscript has been judged scientifically suitable for publication and will be formally accepted for publication once it meets all outstanding technical requirements.

Kind regards,

John P. Hart, Ph.D.

Academic Editor

PLOS ONE

Additional Editor Comments (optional):

Reviewers' comments:

Reviewer's Responses to Questions

**Comments to the Author**

1. If the authors have adequately addressed your comments raised in a previous round of review and you feel that this manuscript is now acceptable for publication, you may indicate that here to bypass the “Comments to the Author” section, enter your conflict of interest statement in the “Confidential to Editor” section, and submit your "Accept" recommendation.

Reviewer #1: All comments have been addressed

Reviewer #2: All comments have been addressed

2. Is the manuscript technically sound, and do the data support the conclusions?

Reviewer #1: Yes

Reviewer #2: Yes

3. Has the statistical analysis been performed appropriately and rigorously? 

Reviewer #1: Yes

Reviewer #2: Yes

4. Have the authors made all data underlying the findings in their manuscript fully available?

Reviewer #1: Yes

Reviewer #2: Yes

5. Is the manuscript presented in an intelligible fashion and written in standard English?

Reviewer #1: Yes

Reviewer #2: Yes

6. Review Comments to the Author

Reviewer #1: The authors have done a great job revising the manuscript. It is now more robust than the first version. This study is a great contribution to bioarchaeology and bone biology more broadly.

Thanks for the opportunity to review this ms.

Justyna Miszkiewicz, PhD

Australian National University

Reviewer #2: I appreciate the authors' effort to correct the manuscript and the diligence with which they responded to individual comments and suggestions. After reviewing the current version of the manuscript, I assess it positively, and I am convinced that it can be accepted for publication.

7. PLOS authors have the option to publish the peer review history of their article (what does this mean?). If published, this will include your full peer review and any attached files.

Reviewer #1: **Yes: **Justyna Miszkiewicz

Reviewer #2: No

---

## [Editor Report · Acceptance letter]

26 Sep 2022

PONE-D-21-33496R1 

Social stratification reflected in bone mineral density and stature: Spectral imaging and osteoarchaeological findings from medieval Norway 

Dear Dr. Brødholt:

I'm pleased to inform you that your manuscript has been deemed suitable for publication in PLOS ONE. Congratulations! Your manuscript is now with our production department. 

Kind regards, 

on behalf of

Dr. John P. Hart 

Academic Editor

PLOS ONE